# Role of Soluble Cytokine Receptors in Gastric Cancer Development and Chemoresistance

**DOI:** 10.3390/ijms26062534

**Published:** 2025-03-12

**Authors:** Francesca Lospinoso Severini, Geppino Falco, Tiziana Notarangelo

**Affiliations:** 1Laboratory of Preclinical and Translational Research, IRCCS CROB Centro di Riferimento Oncologico della Basilicata, 85028 Rionero in Vulture, PZ, Italy; 2Department of Biology, University of Naples Federico II, 80138 Napoli, NA, Italy; 3Biogem, Istituto di Biologia e Genetica Molecolare, 83031 Ariano Irpino, AV, Italy

**Keywords:** gastric cancer, drug resistance, soluble receptors, cytokines

## Abstract

Gastric cancer is among the top five most important malignancies in the world due to the high burden of the disease and its lethality. Indeed, it is the fourth most common cause of death worldwide, characterized by a poor prognosis and low responsiveness to chemotherapy. Multidrug resistance limits the clinical management of the patient. Among these, the role of chronic activation of inflammatory pathways underlying gastric tumorigenesis should be highlighted. Furthermore, the gastric immunosuppressive TME influences the response to therapy. This review discusses the role of soluble cytokine receptors in the development and chemoresistance of gastric cancer, considered as a molecular marker and target of strategies to overcome resistance.

## 1. Introduction

Gastric cancer (GC) represents the fifth most common cancer, ranking as the fourth leading cause of cancer-related death worldwide [1,2]. The GLOBOCAN 2020 [1] database estimated that there were 1,089,103 newly diagnosed GC cases in 185 countries. Based on anatomical locations, GC is classified into cardia GC and noncardia GC [3]. Lauren’s classification divides GC into three groups: diffuse, intestinal, and mixed types [4]. There is a significant difference in GC incidence with sex distribution, approximately 2-fold higher in males than females, and increases with advancing age (>50). Furthermore, geographic variations in the age-standardized incidence rate (ASIR) showed that GC incidence is highest in Asia, followed by Latin America and the Caribbean, Europe, and Oceania, and it is lowest in Africa and North America [5]. GC incidence correlated with the human development index (HDI), depending on four key dimensions of human development: life expectancy at birth, mean years of schooling, expected years of education, and GNI per capita [3]. Countries with very high HDI have higher mortality rates, and countries with medium-low HDI have lower mortality rates [6].

Based on clinical-pathological characteristics, it is possible to identify early gastric cancer (EGC) with a more favorable prognosis and advanced gastric cancer (AGC) characterized by a poor survival rate [3,4,5]. Histologically, EGC is defined as a carcinoma with limited invasion of the mucosa (type M) or submucosa (type SM) of the gastric wall [6]. GC is frequently diagnosed at the advanced or metastatic stages because patients with early-stage GC are mostly asymptomatic and the low sensitivity, and the high rate of false positives of the currently available serum tumor markers (such as CEA, CA19-9, and CA72-4) for early-stage disease (<35%) precludes their use in screening and early diagnosis [7,8,9]. Furthermore, the human epidermal growth factor receptor (HER-2) is used as a marker of eligibility for the first-line treatment with trastuzumab [7] and exhibits a marked intratumor heterogeneity, being heterogeneously distributed in both the primary tumor and metastases [8]. Indeed, the high inter- and intra-tumoral histopathological and molecular heterogeneity leads to limited clinical benefit for patients undergoing conventional chemotherapy [5,10,11,12]. Indeed, the main treatment method for AGC is chemotherapy with a 5-year survival rate of no more than 7%. Hence the need to identify new molecular markers predictive/diagnostic.

## 2. Triggers of GC Development and Risk Factors Associated

Gastric cancer is a multifactorial disease associated with three main triggers: oncogenic infection, genetic susceptibility, and environmental factors [9,10]. Approximately 70% of GC are sporadic, in 10% of cases, familial clustering was observed, and approximately 3% of gastric cancer cases arise in the setting of hereditary diffuse gastric cancer (HDGC), frequently associated with germline mutations in the chromodomain helicase DNA binding protein 1 (CDH1) gene [11]. CDH1 influences cell–cell adhesion mediated by E-cadherin by interfering with processes that regulate cell division, such as the orientation of the mitotic spindle [11]. One of the most important causative etiologic factors for the onset of GC is the gram-negative bacterium H. Pylori, a well-known carcinogen defined as class 1 because of its role in the development of gastric cancer [12]. H. pylori could bind tightly to the gastric epithelial cell layer by bacterial surface proteins (adhesins) by interfering with several major host signal transduction pathways that cause, in turn, cell vacuolar degeneration via the cytotoxin-associated gene A (CagA) and vacuolating cytotoxin A (VacA) proteins [12]. The genetic information exchanged between different strains of H. Pylori following a mixed infection, known as horizontal gene transfer, is very frequent; for this reason, the H. Pylori genome can vary continuously during the colonization of a host. H. pylori is capable of persistently infecting the human stomach, causing a chronic inflammatory response of variable magnitude depending on the host’s genetic signatures; this chronic inflammation stress then induces the oncogenic transformation [13]. In this scenario, the literature reported that interleukin-1-beta (IL-1β) gene cluster polymorphisms, which is an important pro-inflammatory cytokine and a powerful inhibitor of gastric acid secretion, are associated with an increased risk of both hypochlorhydria induced by H. pylori and gastric cancer. Host genetic factors that affect IL-1β may determine why some individuals infected with H. pylori develop gastric cancer, while others do not [13].

In addition, gastric cancer is frequently associated with the Epstein–Barr virus (EBV) infection. It is estimated that 8% of gastric cancer patients carry the EBV genome [14]. After primary infection, EBV establishes a lifelong latency infection in memory B cells, where the virus resides without any consequence in the majority of individuals [15]. However, the virus can evade immune system recognition and lead to the development of different tumors, such as lymphoproliferative disorders, Hodgkin’s lymphoma, Burkitt’s lymphoma, nasopharyngeal carcinoma, and, in particular, gastric cancer [15]. EBV-positive associated GC defines a subtype of tumors characterized by singular clinical and pathological features and a better prognosis among all five adenocarcinomas. Furthermore, EBV-positive gastric adenocarcinomas have the highest immunogenicity, and GC could be an applicable candidate for further immunotherapy investigation in GC [9,10].

## 3. Clinical Management

Cytotoxic chemotherapy is the first-line treatment of advanced gastric cancer (AGC), yet responses are heterogeneous, and chemoresistance represents a significant limitation responsible for unsatisfactory clinical outcomes and the high mortality rate of GC patients [16]. The first-line treatment is the association of platinum compounds (cisplatin/oxaliplatin), plus a fluoropyrimidine, combined with trastuzumab in patients that are HER2 positive [7]. The second-line treatments are planned on the use of ramucirumab, a monoclonal antibody directed against vascular endothelial growth factor receptor 2 (VEGFR2), or the taxanes alone or in combination with ramucirumab or in association with irinotecan or docetaxel [16,17]. The third-line treatment is based on the use of TAS 102 (trifluridine-tipiracil) [18,19]. Furthermore, in recent years, programmed cell death protein 1 (PD-1) or programmed death-ligand 1 (PD-L1) inhibitors have been used in the management of advanced gastric cancer treatment [20]. Recently, phase III clinical trials have led to FDA approval for the use of immune checkpoint inhibitors (ICIs), such as nivolumab or pembrolizumab with standard first-line treatments (5-fluoropyrimidine and platinum compound), improving the clinical response [21]. Indeed, results of the CheckMate-649 trial showed a greater effectiveness of nivolumab in combination with chemotherapy in first-line treatment [22]. Similarly, the same findings are observed in the KEYNOTE-811 trial, which demonstrated the efficacy of first-line pembrolizumab in combination with trastuzumab and chemotherapy in HER2-positive patients [23]. Unfortunately, although the management of advanced gastric cancer (AGC) has gradually improved due to the resistance mechanisms that are established, patients have a median overall survival (OS) of less than 12 months [21,24].

## 4. Drug Resistance in Gastric Cancer

Therapy resistance reduces the effectiveness of many chemotherapies currently used to treat cancer. In oncology, it is the main cause responsible for therapeutic failure at the beginning of the treatment or after an initial response to chemotherapy. It is a complex multifactorial phenomenon involving several interconnected or independent pathways that cause changes within cancer cells that become unresponsive to the treatment [25,26]. Therapy resistance represents a major problem when treating cancer patients because cancer cells develop mechanisms that represent a limit on the effect of therapeutic agents, leading to more aggressive clones that contribute to poor prognoses. The resistance can be primary or intrinsic and secondary or acquired if it occurs, respectively, before or after exposure to a chemotherapy drug. Indeed, the multidrug-resistant (MDR) phenotype is associated with intrinsic resistance to a broad spectrum of drugs and correlates with an aggressive phenotype and negative follow-up [25,26,27]. The potential mechanisms of MDR currently can occur due to a variety of mechanisms, such as increased drug inactivation, drug efflux from cancer cells, enhanced repair of chemotherapy-induced damage, activation of pro-survival pathways, and inactivation of cell death pathways [28]. The altered drug efflux mechanisms involve an aberrant expression both of P-glycoprotein (P-gp1), belonging to the ABC transporters family that extrudes many types of drugs from cancer cells, and glutathione S-transferases (GSTs), involved in the detoxification pathway, that catalyze the conjugation of glutathione (GSH) to compounds for easy elimination [28,29]. P-gp1, on the other hand, can export glutathione (GSH), glucuronate, or sulfate conjugates of organic anions, which is induced by cyclooxygenase-2 (COX-2) and involved in inflammation and often associated with resistance to anticancer drugs [30,31]. Among the mechanisms associated with resistance, the tumor microenvironment (TME) dictates the therapy’s efficacy, representing a probable therapeutic target [32]. TMEs include CD4 helper T (Th) cells, CD8 cytotoxic T (CTL) cells and regulatory T (Treg) cells, effector T and B lymphocytes, NK cells, dendritic cells (DCs), tumor-associated macrophages (TAMs), myeloid-derived suppressor cells (MDSCs), endothelial cells (ECs), cancer-associated fibroblasts (CAFs), and extracellular matrix (ECM) components, as well as soluble products, such as chemokines, cytokines, growth factors, and extracellular vesicles [33]. Indeed, TME, through the secretion of high levels of inflammatory cytokines, activates the inflammatory pathways promoting tumor progression, the activation of angiogenetic mechanisms, and the evasion of apoptotic pathways that confer a survival advantage [18,19,20].

## 5. Role of Cytokines in Gastric Cancer Resistance and Development

Inflammatory mechanisms dictate tumor development and response to therapy, which either promotes or suppresses tumor progression; chronic inflammation facilitates tumor progression and treatment resistance, while the induction of acute inflammatory reactions often stimulates the immune response, leading to antitumor activity [34]. The progression of chronic inflammatory diseases could depend on the persistence of immune and inflammation responses. The gastric TME consists of high levels of several inflammatory cytokines [35,36]. The strict correlation between the onset of cancer and chronic inflammation is becoming increasingly stronger [34,37,38]. TME establishes a reciprocal interaction between the tumor cells immersed in a network of stromal cells and cells of innate and adaptive immunity that ensures the maintenance of the malignant properties of tumor cells and controls cancer progression [39]. Interestingly, the distinctive features of TME, which can also induce significant genetic changes in cancer cells, develop early in solid tumors [40]. Within the immune tumor microenvironment, cytokines are relevant as they can act on a wide range of cell types, including the gastrointestinal epithelium, and create a dense communication network that branches even at sites distant from the origin of the tumor. These molecules, including interleukins, interferons, tumor necrosis factors, and chemokines, bind to receptors and trigger intracellular signaling pathways to modulate gene transcription through different transcription factors, such as activator protein 1 (AP-1), mitogen-activated protein kinases (MAPKs), and nuclear factor kappa B (NF-κB), that can also directly or indirectly influence epigenetic regulations [41,42]. In GC, several inflammatory cytokines have been evaluated to determine if they can be applied as prognostic biomarkers. Several pro-tumor cytokines could be used as markers of GC development and progression [43,44,45,46,47]. IL-4 regulates the immune response and, together with IL-13, is involved in the crosstalk with the TME by activating TAM and MDSCs, mediating tumor progression and relapse [48]. Furthermore, IL-4 and IL-13 reduce the chemotherapy sensitivity to cisplatin in gastric cancer cells [49]. IL-6 and IL-11 activate the JAK-STAT3 pathway in GC cells. Il-6 is a well-known marker of poor prognosis and low responsiveness to platinum-based compounds (neoadjuvant chemotherapy) [35]. IL-11 contributed to resistance to cisplatin through gp130/JAK/STAT3/Bcl2 pathway activation [50]. Similarly, IL-33 plays a role both in resistance to cisplatin and promotes cell invasion activating the JNK pathway [51]. IL-33 is also associated with the depth of invasion distant to metastasis at an advanced stage (stage III/IV) [52]. IL-8 is an inflammatory signaling molecule, secreted in gastritis and associated with the risk of neoplastic transformation [53]. In addition, in GC cells, increased IL-8 expression correlated with more invasion/migration mechanisms and oxaliplatin resistance by stimulation of NF-κB and AKT axis [54]. IL-1β is an important mediator of the inflammatory response crucial for the progression of GC [55]. Furthermore, IL-24 reduces the response to cisplatin through a mechanism involving the upregulation of Bax and downregulation of P-gp1 and Bcl-2 [56]. Moreover, IL-10, IL-17, and TNFα represent markers of bad a prognosis [57,58,59,60]. Similarly, the chemokines CXCL1, CXCL5, CXCL11, and CXCL14 correlate to the presence of distant metastasis and nodal involvement and are associated with relapse [61,62,63,64,65]. Exploring the roles of these molecules in tumor biology and immune response has opened new challenges regarding the targeting of cytokine signaling pathways for cancer therapy, exploiting the dual nature of cytokines as promoters and suppressors of tumorigenesis [66]. For example, for their anti-tumor activity, IFN-α and IL-2 have been approved by the Food and Drug Administration (FDA) for the treatment of various cancers (melanoma, follicular lymphoma, hairy cell leukemia, acquired immunodeficiency syndrome (AIDS)-associated Kaposi’s sarcoma, and renal cell carcinoma) [67,68]. Currently, in the online clinical trial registry clinicaltrials.gov, there are 16 clinical trials targeting cytokines for gastric cancer therapy, of which 14 studies are interventional and 2 observational.

## 6. Soluble Receptor Intervention in Cytokine Signaling Pathways

Soluble receptors are soluble forms of receptors found in the extracellular space with a pivotal role in cellular signaling and disease pathogenesis. Receptors include a cytoplasmic domain, a transmembrane domain, and an extracellular domain. Soluble receptors binding its ligand could influence signaling pathways [69,70]. These receptors represent a likely biomarker for early diagnosis as they correlate with disease severity and allow minimally invasive serum assay. Soluble receptors are involved in cancer progression, metastasis, and immune escape. While their clinical relevance to cancer has become increasingly evident, their therapeutic use remains poor. Nevertheless, targeting soluble receptors could represent excellent therapeutic changes [71].

As previously mentioned, several soluble cytokine receptors and immune checkpoints in the tumor microenvironment are involved in cancer progression and immune evasion in response to therapy [72]. Cytokines orchestrate immune responses modulating cellular functions, including cell proliferation, differentiation, and migration. The receptors bind their respective cytokines, initiating a series of intracellular signaling cascades. Numerous studies have reported that their dysregulation is closely associated with the pathogenesis of inflammatory diseases and cancer. Indeed, prolonged activated or suppressed signaling of several cytokines could favor immune evasion in the tumor microenvironment and cancer development. Furthermore, tumor cells could produce some cytokines and their receptors through an autocrine loop that further enhances cell survival and proliferation [69,70].

The regulation and complexity of the intracellular signaling pathways are enriched by several ligands, receptors, and co-receptors on the cell surface and in the extracellular space. Cytokine receptors are found in the membrane-bound form and soluble forms. Soluble forms of cytokine receptors can be released into the extracellular environment. It is important to underline that the levels of soluble cytokine receptors have been reported to be higher in the serum of patients with various cancers than in healthy controls.

In this scenario, the soluble receptors should be emphasized due to their unique characteristics that make them promising prognostic and diagnostic markers in various diseases, including cancer [66,67]. Soluble receptors can be measured easily and non-invasively in a patient’s serum, retain their high binding affinity with the ligand, can reach target tissues even from a distant injection or production site, and can often mediate their effects with a pleiotropic action on different cell types than that of classical receptors. Soluble cytokine receptors are created through several mechanisms: (i) proteolytic cleavage of an existing membrane receptor (ectodomain-shedding), (ii) synthesis of an mRNA lacking the transmembrane domain sequence, via alternative mRNA splicing or the use of intronic alternative polyadenylation (PAS) sites, and (iii) release of transmembrane receptors through extracellular vehicles, such as exosomes [71]. Ectodomain shedding is a process in which transmembrane proteins exposed on the cell surface or cellular organelles are proteolytically cleaved and released by enzymes, called “sheddases”. The cleaved extracellular domain (ectodomain) of a membrane-bound receptor is released into the extracellular space and transported in a soluble form. The key enzymes in ectodomain shedding are the well-known ADAM family (Figure 1) [73]. They consist of a catalytic metalloproteinase domain that functions in shedding, a disintegrin domain, a cysteine-rich domain, a transmembrane domain, and a C-terminal cytoplasmic domain that is involved in activity regulation. The short C-terminal fragment (CTF) that remains at the plasma membrane because of receptor cleavage is further processed by the γ-secretase protease complex to release the intracellular domain (ICD) fragment. Among the ADAM family, ADAM10 and ADAM17, which have similar structures, are directly involved in the pathogenesis of different tumors [74,75]. ADAM10 and ADAM17 modulate tumor progression through their influence on several distinct cellular pathways. Indeed, they affect cell–cell interactions and cell migration, releasing cell adhesion molecules. In addition, ADAM10 and ADAM17 are involved in the shedding of EGFR ligands regulating, in turn, the activation of the EGFR tyrosine kinase family [76]. In gastric cancer patients, ADAM10 and ADAM17 are associated with cancer progression and are biomarkers of poor prognosis [77,78]. Cytokine receptors cleaved by sheddases include class I cytokine receptors (e.g., IL-2 receptor and IL-6 receptor), the tumor necrosis factor (TNF) receptor superfamily, and the IL-1 receptor/Toll-like receptor superfamily [77,79].

## 7. Protumor Cytokines in Gastric Cancer and Their Soluble Receptors

The prolonged or suppressed signaling of one or more cytokines may promote immune evasion in the tumor microenvironment, favoring the development of cancer. Indeed, the characterization of the regulatory properties of soluble receptors needs to be investigated [40,72].

### 7.1. Soluble IL-6R (sIL-6R) and IL-11R (sIL-11R)

The interleukin-6 (IL-6) is critical for several physiological processes, such as inflammation and cancer. The IL-6 receptor is present in two forms: membrane-bound IL-6R and the soluble form. While the IL-6R receptor is expressed only in immune cells, the GP130 co-receptor has ubiquitous expression. In classic signaling, IL-6 binds to membrane-bound IL-6R, inducing homodimerization of the signal transducer protein GP130 and activation of signaling. The soluble form of the IL-6 receptor retains its ability to bind to IL-6, forming the IL-6/sIL-6R complex and acting as a ligand transporter protein with an agonist role compared to classical signaling [80].

When IL-6 is bound to IL-6R, all cells expressing the gp130 co-receptor become sensitive to IL-6. This type of signaling by the soluble receptor is called trans-signaling and acts as a signal amplifier. IL-6 signaling is one of the most complex pathways, as there is a soluble form of the gp130 co-receptor (sgp130) that can bind as a bait to the IL-6/Sil-6R complex preventing its binding to gp130, effectively blocking trans-reporting [80,81]. Two isoforms of sIL-6R, DS-sIL-6R, and PC-sIL-6R were discovered, respectively, generated by alternative mRNA splicing and proteolytic processing. The two isoforms differ by a proximal COOH-terminal sequence (GSRRRGSCGL) generated during splicing [82,83]. Although little is known about the expression of these two isoforms, their release is differentially regulated. ADAM17 is the enzyme that mainly mediates IL-6R proteolysis on the cell membrane [84]. sIL-6R is present in the plasma of healthy individuals (∼25–35 ng/mL) and elevated levels of this soluble receptor have been detected in numerous disease states, including cancer [85]. Similar to sIL-6, the soluble IL-11 receptor (sIL-11R) could associate with the gp130 sub-units bound to the membrane to carry out signal transduction. Unlike sIL-6R, a unique isoform of sIL-11R has been discovered that is generated through proteolytic processing by the enzymes ADAM9 and ADAM10 [86]. It is known that IL-6 is a mediator of tumorigenesis in several human malignancies [24,25,26], being overactive in many tumor types, including GC, and that the aberrant activation of the JAK/STAT pathway being dependant on IL-6 dictates development and resistance in GC. High levels of IL-6 are correlated to fluorouracil and cisplatin resistance [80]. Furthermore, IL-6 is strictly related to immune responses affecting tumor progression. Indeed, gastric cancer mesenchymal stem cells (GC-MSCs) and neutrophils interact through the IL-6-STAT3-ERK1/2, signaling axis, stimulating neutrophil polarization in the N2 phenotype in gastric cancer [87]. The strong inflammatory state generated by MDSCs is strictly linked to the recruitment of inflammatory cytokines, such as IL-1β, TNF-α, IL-6, and IL-8, and chemokines, such as CXCL-1 and CXCL-12 [88,89]. Furthermore, CAFs secrete IL-6 and IL-8, enhancing resistance to chemotherapy through the activation of the JAK/STAT axis [90]. TAMs and CAFs cooperate through an intricate network involving different cytokines and other factors with the activation of a cascade of pathways that culminate in drug resistance [91]. In the view of the significance of IL-6/sIL-6R trans-signaling in tumor progression, targeting this trans-signaling has therapeutic potential.

### 7.2. Soluble IL-1R (sIL1R-II)

The IL-1R family includes ten transmembrane proteins with a similar structure, consisting of an extracellular portion generally composed of three Ig-like domains in the N-terminal portion (D1, D2, and D3), usually responsible for a ligand, a transmembrane domain, and an intracellular portion, with the Toll-IL-1-receptor (TIR) domain important for initiating signal transduction [92]. Only the IL-1R1 and IL-1R2 receptors are capable of binding IL-1α and β that belong to the IL-1 family. IL-1R2 receptor occurs in both soluble and membrane-bound forms, has a very short intracellular portion lacking the TIR domain, and consequently is unable to activate signaling. Both membrane-bound IL-1R2 and sIL-1R2 can capture IL-1, and both prevent it from binding to the IL-1R1 signaling receptor. The complex of metalloproteinase TACE/ADAM17 and the aminopeptidase ARTS-1358-361 or α- and β-secretase generates the soluble form of IL-1R2 (sIL-1R2) [93]. Regulation of the activity of the proinflammatory cytokine IL-1 is complex, involving the accessory protein (AcP) that adds another layer of complexity to the regulation of the IL-1 action, increasing the affinity of binding between IL-1α and IL-1β and the soluble form of the IL-1R2. Overall, AcP contributes to the antagonism of the IL-1 action by IL-1R2 [92]. Indeed, IL-1R2 acts by modulating the IL-1 availability, acting as a negative regulator of the IL-1 system for the signaling receptor [92]. IL-1R2 has important clinical significance since it is abnormally expressed in many human inflammatory diseases and cancers. Studies revealed that in GC, highly expressed IL-1R2 correlates negatively with overall survival, indicative of increased levels of IL-1R2 being involved in the initiation and progression of GC [94]. It is important to highlight the role of IL-1β as an important mediator of the inflammatory response produced by activated macrophages, monocytes, and a subset of dendritic cells linked to high-grade mucosal inflammation that is crucial for the progression of GC. Furthermore, IL-1β could recruit other inflammatory cytokines, activating immune cells within the gastric mucosa [55,95]. In this view, IL-1R2 could be employed to develop therapeutic approaches targeting GC.

### 7.3. Soluble TNF (sTNF)

Tumor necrosis factor (TNF) is a potent proinflammatory cytokine, released mainly by stimulated macrophages involved in many inflammatory diseases. TNF binds two receptors, TNF receptor 1 (TNFR1), involved in inflammation mechanisms, and TNF receptor 2 (TNFR2), involved in survival and proliferation pathways, which have similar extracellular structures. The soluble form of TNF is generated through proteolytic cleavage (mediated by ADAM), synthesis via alternative mRNA splicing, or release in extracellular vesicles [69]. The literature data reported increased serum concentrations of soluble TNFR1 in patients with advanced gastric, suggesting that the receptor could be an important prognostic factor in predicting the severity of GC [96]. Furthermore, high sTNF-R2 levels are associated with a significant increase in overall mortality [97]. It is known that TNF is responsible for the activation of the Treg subtype in GC and the consequent GC progression through the e MAPK signaling pathway [71,98].

### 7.4. Soluble IL-33R (sSt2)

Suppression of tumorigenicity 2 (ST2), also known as interleukin-1 receptor-like 1 (IL1RL1), is one of the natural receptors of IL-33. Three major isoforms, ST2L (transmembrane form), sST2 (soluble form), and ST2V, are generated by alternative splicing. IL-33 acts through a cell surface receptor complex, ST2 (IL-1 receptor-like 1, IL1RL1) and IL1RAcP (IL-1 receptor accessory protein) induce cell proliferation and activate inflammatory pathways in immune cells [99]. This clustered domain recruits signaling adapters and kinases to activate transcription factors (ERK, NF-kB, p38, and JNK) in tumor cells and generate a protumoral inflammatory TME [100]. The soluble form of the IL-33 receptor, sST2, is generated by alternative activation of the suppression of tumorigenicity (ST) 2 gene promoter. It is reported that the transcription factor GATA-binding factor 2 (GATA2) regulates the transcription of the ST2L gene by activating the distal promoter, and GATA 3 regulates the transcription of the sST2 gene by activating the proximal promoter [101]. Currently, there is no single front of data that clarifies its function. In particular, the research carried out in the field of cancer is discordant. We can say that, in general, sST2 is described as a decoy receptor that limits the availability of IL33. In some types of cancer such as gastric and breast cancer, sST2 is assigned an agonist role compared to classical signaling [102,103]. Similarly, IL-33, a member of the IL-1 family, represents a potent activator of Th2 immunity. IL-33 is identified as a key promoter in several inflammatory disorders and immune-mediated pathological conditions [100]. In GC, high expression correlated with poor prognosis, invasion, and metastasis. IL-33 is also associated with resistance to apoptosis induced by platinum-containing chemical compounds via the MAPK/JNK1 pathway [51]. Furthermore, IL-33, a member of the IL-1 family, has the ability to potently activate Th2 immunity [104]. In gastric TME, IL-33 is involved in the progression through activation of mast cells, which in turn attract TAMs. IL-33 mediates interactions between CAFs and GC cells. In particular, CAFs secrete high levels of IL-33, which activates the ERK1/2-SP1 signaling pathway in cancer cells and the expression of the EMT-associated transcription factor, ZEB2 [105]. Several studies suggest that soluble ST2 is associated with advanced and metastatic disease in GC patients making it a valuable biomarker of gastric cancer progression and pathogenesis [106,107,108].

## 8. Targeting Soluble Receptors for Cancer Therapy

Soluble receptors represent probable biomarkers for cancer diagnosis and prognosis and could be used as therapeutic targets in cancer treatment. Direct targeting of soluble receptors or their pathways might be an effective therapeutic strategy that could enhance the efficacy of chemotherapy [80]. Small-molecule compounds or monoclonal antibodies that directly target several soluble receptors have been developed, such as tocilizumab (directed against IL-6) [83]. However, these antibodies are not capable of distinguishing the soluble receptor from the bound counterpart. An alternative inhibitor that can be considered is olamkicept, a soluble gp130–Fc fusion protein (sgp130Fc), which is a fusion protein of the extracellular portion of gp130 and the Fc region of a human IgG1 antibody [109]. Another therapeutic strategy could be the inhibition of enzymes involved in the shedding of membrane-bound receptors. The literature data have shown that ADAM expression levels are increased in several tumors, including gastric cancer [73,75,76,77,78]. Preclinical studies have reported that ADAM modulation inhibits tumor cell migration, invasion, and growth [110]. Moreover, since serum levels of soluble immune checkpoints are correlated with resistance to immunotherapy, targeting soluble immune checkpoints might be beneficial for immunotherapy-resistant cancer patients [111]. Therefore, combination therapy with existing therapeutic strategies and inhibition of soluble receptors could represent a solution to overcome the limitations of current treatments.

## 9. Conclusions

Soluble cytokine receptors proved to be a crucial feature in cancer progression and therapy efficacy. Here, we highlighted the role of the soluble receptors, which form a ligand–receptor complex with their cytokines to stimulate proliferation, invasion, and drug resistance mechanisms. In this scenario, soluble cytokine receptors are proposed as a valid, minimally invasive, diagnostic tool for early diagnosis and for monitoring the clinical response. In recent years, new therapeutic strategies targeting specific cells within the tumor microenvironment have been developed. Identifying relevant biomarkers is required to recognize patients who are expected to benefit from therapy [112]. Similarly, in other tumors, multiple inhibition strategies are effectively used, and drugs targeting the molecular microenvironment will probably require a combination with other agents or approaches in the clinical practice [15,113,114,115,116]. Indeed, tumors often present mutations in multiple genes, so it is unlikely that the use of a single molecularly targeted drug will be clinically effective [115,117]. Therefore, it has become necessary to develop multiple targeted therapies capable of acting simultaneously on multiple molecular targets to obtain optimal inhibition of tumor growth. In this scenario, targeting soluble receptors with monoclonal antibodies or small molecule inhibitors could represent a useful strategy to enhance the limitations of current treatments. In the future, it could prove successful in the fight against tumor diseases, as it could allow a specific, selective, and increasingly tailored therapy [32,33,118].

## Figures and Tables

**Figure 1 ijms-26-02534-f001:**
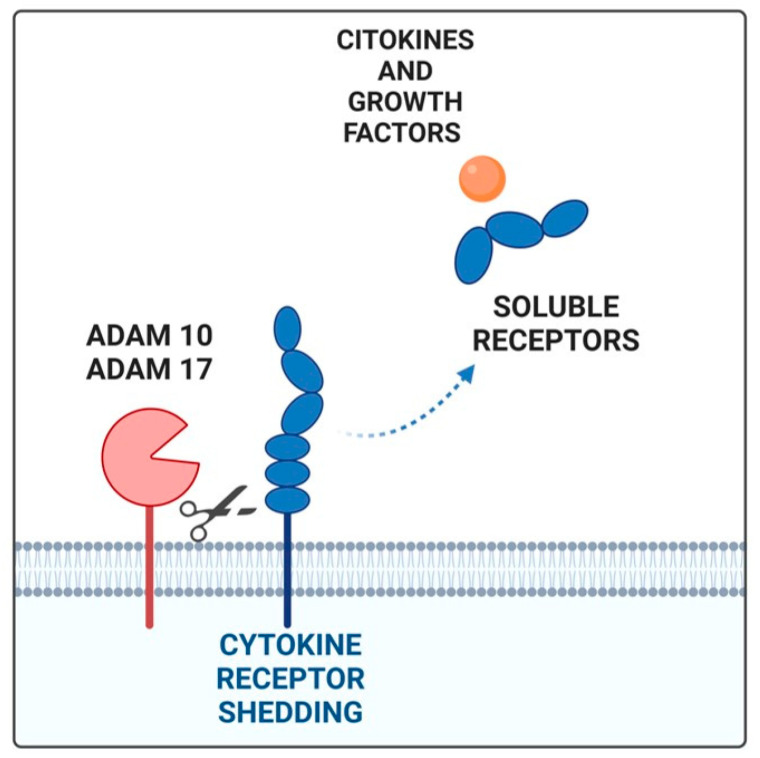
The soluble receptor generated by shedding (blue) by ADAM17 and ADAM10 proteases binds its ligand (orange). Created by Biorender.com (accessed on 27 January 2025).

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
