# Peer review of "Role of Soluble Cytokine Receptors in Gastric Cancer Development and Chemoresistance"

_ijms, 2025, doi:10.3390/ijms26062534_

Round 1
Reviewer 1 Report
Comments and Suggestions for Authors
From a scientific perspective, the Review presents an interesting and significant contribution to the field. However, in its current form, it cannot be accepted. The authors should revise the text to enhance clarity and improve its structure. English language use requires refinement to communicate the research more effectively . Several minor revisions are necessary, as outlined below.
As written in the “Instructions for Authors”, references must be numbered sequentially according to their appearance in the text and listed individually at the end of the manuscript. In the text, reference numbers should be placed in square brackets [ ], and placed before the punctuation; for example [1], [1–3] or [1,3]. In this Review, the references are not listed in order of appearance, and square and round brackets are used interchangeably. Furthermore, there are inaccuracies that require correction. An error in the insertion of two citations has led to the incorrect scaling of all subsequent references:“45. Patel, P.; Chatterjee, S. 46. Lee, H.-M.; Lee, H.-J.; Chang, J.-E. Inflammatory Cytokine: An Attractive Target for Cancer Treatment. Biomedicines 2022, 10, 2116, https://doi.org/10.3390/biomedicines10092116.” At page 6, paragraph “6.3 Soluble IL-1R (sIL1R-II)” there is a references numbered 91 that does not exist. Frequently, the period is omitted at the end of a sentence following a reference. A thorough review of punctuation is necessary.
The pages numbering is incorrect, with the numbering restarting for both the table and references sections.
Acronyms, abbreviations and initialisms should be defined the first time they appear in each of three sections: the abstract, the main text, and figure/table/scheme captions. Subsequently, they should be used consistently as acronyms throughout the text.
Italics must be used for the genus and species when using Latin nomenclature of organisms. The first time the name is used, it should be spelled out in full, but for further uses, it can be abbreviated to the first letter. It is essential to standardize both the use of italics and the abbreviation of the first letter throughout the text (e.g., H. Pylori).
The structure of numerous sentences is excessively complex, with sentences that are overly long and do not adhere to a logical progression. The result will be that readers are confused, especially if the sentence starts on one theme, adds a lengthy subclause for explanation, then goes back to the original theme.
As written in the “Instructions for Authors”, the structure of Reviews can include an Abstract, Keywords, Introduction, Relevant Sections, Discussion, Conclusions, and Future Directions. As currently written, this review appears to lack a crucial component. It may be beneficial for the reader if a final paragraph of discussion/conclusion is added, summarizing the main topics and discussing potential future directions.
At page 4, the phrase from ”distant metastasis and advanced stage (stage III/IV) [53]. IL-8” it’s written in a different font size. Please ensure that consistent typography has been employed throughout the text.
Please verify that all verbs have been used in the correct tense (e.g. At page 3 “Furthermore, IL-4 and IL-13 reduces the chemotherapy sensitivity to cisplatin in gastric cancer cells [50].”).
Figure 1 could be enhanced in terms of image quality and textual clarity.
The table should be revised and standardized from a typographical perspective. Occasionally, words are underlined or bolded, but a consistent format is not maintained throughout the table. Additionally, it would be beneficial to include citations within the table to improve comprehension.
Check the orthography, as there are occasional spelling errors in the text. (e.g. In Table 1 “*IL-6 and TNF-α secreted by macrophages induce PD-L1 expression in turmoral cells via the NF-κB and STAT3 signaling pathways”).
Comments on the Quality of English LanguageAs stated above, the authors should revise the manuscript to enhance sentence clarity and coherence, facilitating the reader's comprehension of the scientific topic. The quality of the English language must be improved, as there are multiple inaccuracies in spelling, verb tense, and punctuation.
Author Response
Reviewer 1
From a scientific perspective, the Review presents an interesting and significant contribution to the field. However, in its current form, it cannot be accepted. The authors should revise the text to enhance clarity and improve its structure. English language use requires refinement to communicate the research more effectively. Several minor revisions are necessary, as outlined below.
As written in the “Instructions for Authors”, references must be numbered sequentially according to their appearance in the text and listed individually at the end of the manuscript. In the text, reference numbers should be placed in square brackets [ ], and placed before the punctuation; for example [1], [1–3] or [1,3]. In this Review, the references are not listed in order of appearance, and square and round brackets are used interchangeably. Furthermore, there are inaccuracies that require correction. An error in the insertion of two citations has led to the incorrect scaling of all subsequent references:“45. Patel, P.; Chatterjee, S. 46. Lee, H.-M.; Lee, H.-J.; Chang, J.-E. Inflammatory Cytokine: An Attractive Target for Cancer Treatment. Biomedicines 2022, 10, 2116, https://doi.org/10.3390/biomedicines10092116.” At page 6, paragraph “6.3 Soluble IL-1R (sIL1R-II)” there is a references numbered 91 that does not exist. Frequently, the period is omitted at the end of a sentence following a reference. A thorough review of punctuation is necessary. Done
The pages numbering is incorrect, with the numbering restarting for both the table and references sections. Done
Acronyms, abbreviations and initialisms should be defined the first time they appear in each of three sections: the abstract, the main text, and figure/table/scheme captions. Subsequently, they should be used consistently as acronyms throughout the text. Done
Italics must be used for the genus and species when using Latin nomenclature of organisms. The first time the name is used, it should be spelled out in full, but for further uses, it can be abbreviated to the first letter. It is essential to standardize both the use of italics and the abbreviation of the first letter throughout the text (e.g., H. Pylori).
Done
The structure of numerous sentences is excessively complex, with sentences that are overly long and do not adhere to a logical progression. The result will be that readers are confused, especially if the sentence starts on one theme, adds a lengthy subclause for explanation, then goes back to the original theme.
As written in the “Instructions for Authors”, the structure of Reviews can include an Abstract, Keywords, Introduction, Relevant Sections, Discussion, Conclusions, and Future Directions. As currently written, this review appears to lack a crucial component. It may be beneficial for the reader if a final paragraph of discussion/conclusion is added, summarizing the main topics and discussing potential future directions.
Thanks the review for your suggestion. We added the discussion and conclusion paragraphs in the text.
At page 4, the phrase from ”distant metastasis and advanced stage (stage III/IV) [53]. IL-8” it’s written in a different font size. Please ensure that consistent typography has been employed throughout the text. Done
Please verify that all verbs have been used in the correct tense (e.g. At page 3 “Furthermore, IL-4 and IL-13 reduces the chemotherapy sensitivity to cisplatin in gastric cancer cells [50].”). Done
Figure 1 could be enhanced in terms of image quality and textual clarity. Thank you very much for your comment; we have modified the figure according to your suggestion.
The table should be revised and standardized from a typographical perspective. Occasionally, words are underlined or bolded, but a consistent format is not maintained throughout the table. Additionally, it would be beneficial to include citations within the table to improve comprehension. Thanks for your comment; we have removed the table and simplified it in the text.
Check the orthography, as there are occasional spelling errors in the text. (e.g. In Table 1 “*IL-6 and TNF-α secreted by macrophages induce PD-L1 expression in turmoral cells via the NF-κB and STAT3 signaling pathways”). Done
Comments on the Quality of English Language
As stated above, the authors should revise the manuscript to enhance sentence clarity and coherence, facilitating the reader's comprehension of the scientific topic. The quality of the English language must be improved, as there are multiple inaccuracies in spelling, verb tense, and punctuation.
Thanks for your comments; we check the text with grammarly software
Response 1.
We hope that with the latest corrections made and with the rewriting of the discussion and conclusion. I hope the manuscript has improved enough to be considered publishable.

Reviewer 2 Report
Comments and Suggestions for Authors
- In the introduction part, the author need to include more details about the 1. histological subtypes (intestinal vs. diffuse), genetic mutations commonly associated with gastric cancer (e.g., CDH1, TP53), and the role of genomic instability. 2. Explain why cytokines and their receptors are critical in cancer biology, particularly focusing on their dual roles in promoting and inhibiting tumor growth. Introduce soluble receptors as modulators of cytokine signaling. 3. Explain why soluble receptors are being highlighted in this review—what makes them promising targets or biomarkers compared to traditional membrane-bound receptors?
- In Paragraph 3, the mechanisms of multidrug resistance (MDR) need to be elaborated. Include specific examples of drug efflux mechanisms like P-glycoprotein involvement and how it impacts chemotherapy efficacy.
- The author need to address the limitations of current biomarkers (HER2, CEA, CA72-4). Highlight the intratumoral heterogeneity of HER2 and the non-specificity of CA72-4, emphasizing the need for more reliable diagnostic markers.
- In paragraph 5, the author should clarify the role of ADAM proteases in the generation of soluble receptors like sIL-6R and sIL-1R and discuss the implications of soluble receptor shedding in cancer progression and chemoresistance.
- The author needs to add the conclusion and discussion. Summarize some key findings and suggest specific future research directions. Focus on how targeting soluble receptors could translate into clinical therapies for overcoming chemoresistance.
Author Response
Rionero in Vulture, 28/02/2025
Dear Editor,
Thank you very much for the feedback provided and the opportunity to resubmit our manuscript (ijms-3474649) entitled “Role of soluble receptors cytokines in development and chemoresistance of gastric cancer”.
We would like to thank the reviewers for their careful reading of our manuscript. In order to keep the focus of our work, we revised the paper according to the reviewers.
We have addressed all the reviewers’ comments and the responses are appended below.
We hope that in the revised version our manuscript is suitable for publication on International Journal of Molecular Sciences.
Reviewer 2
In the introduction part, the author need to include more details about the
- Histological subtypes (intestinal vs. diffuse), genetic mutations commonly associated with gastric cancer (e.g., CDH1, TP53), and the role of genomic instability. Thanks the review for your suggestion, the text has been revised at several points, placing emphasis on some points of greater relevance to the paper.
- Explain why cytokines and their receptors are critical in cancer biology, particularly focusing on their dual roles in promoting and inhibiting tumor growth. Introduce soluble receptors as modulators of cytokine signaling. Thanks for your comment; we have added this information in the text, according to your suggestion.
- Explain why soluble receptors are being highlighted in this review—what makes them promising targets or biomarkers compared to traditional membrane-bound receptors? Done
In Paragraph 3, the mechanisms of multidrug resistance (MDR) need to be elaborated. Include specific examples of drug efflux mechanisms like P-glycoprotein involvement and how it impacts chemotherapy efficacy. Done
The author need to address the limitations of current biomarkers (HER2, CEA, CA72-4). Highlight the intratumoral heterogeneity of HER2 and the non-specificity of CA72-4, emphasizing the need for more reliable diagnostic markers. Done
In paragraph 5, the author should clarify the role of ADAM proteases in the generation of soluble receptors like sIL-6R and sIL-1R and discuss the implications of soluble receptor shedding in cancer progression and chemoresistance. Done
The author needs to add the conclusion and discussion. Summarize some key findings and suggest specific future research directions. Focus on how targeting soluble receptors could translate into clinical therapies for overcoming chemoresistance.
Thanks the review for your suggestion. We added the discussion and conclusion paragraphs in the text.
Response 2.
We would like to thank you for the feedback. Thank you very much for your comment; we have modified the manuscript according to your suggestion. I hope the manuscript has improved enough to be considered publishable.
Kind regards,
Tiziana Notarangelo, Ph.D.,
Laboratory of Preclinical and Translational Research,
IRCCS-CROB, Referral Cancer Center of Basilicata,
85028, Rionero in Vulture, Italy
Tel. 0972 726239

Reviewer 3 Report
Comments and Suggestions for Authors
1. Should reword the following from the introduction as it is very confusing as is: “Incidence rates are highest in East Asia (Japan and Mongolia) and Eastern Europe and in general a direct correlation has been found between the incidence rate of gastric cancer and the country-level Human Development Index (HDI); index assessing four key dimensions of human development (Life expectancy at birth, mean years of schooling, expected years of schooling, GNI per capita)”
2. Bottom of section 2: “Cytokines may be a macro group of such biochemical markers that need more intensive research” comes out of left field and needs some introduction before making that claim. Or just remove it and talk about it later
3. Page 4 paragraph 1: "Moreover, IL-10, IL-17 and TNFa represent markers of bad prognosis” does not have citations
4. There is a large chunk of text in the first paragraph of 6.1 that does not have appropriate citations (sIL-6R intro information)
5. Should figure 1 be showing an inhibitor? Inhibitors of soluble cytokines haven’t been discussed at this point. Text is “ leads us to hypothesize the role of these vesicles in the amplification of cytokine signaling through fusion with cells that do not normally express such receptors and through the same generation of their soluble receptors” but the figure doesn’t show that at all. Maybe this figure should go at the end?
5.5 - As I continued reading - it doesn't seem like inhibitors of soluble receptors are ever discussed in this paper, so I am unsure why the authors have included them in the figure.
6. There is too much going on in the table - it feels like the text was just rewritten into the table. Should be condensed substantially to make more readable.
7. The paper ends abruptly without any conclusions. I was expecting a section on inhibitors of soluble receptors or at least the impact of soluble receptors on chemoresistance and development of gastric cancer. The manuscript feels more like a review of the role of cytokines in these processes, not of the soluble receptors. An additional section or subsection directly addressing the role of the soluble receptors in gastric cancer is needed.
Comments on the Quality of English LanguageThere are substantial typos throughout as well as grammatical errors that impact the clarity of this paper.
Author Response
Rionero in Vulture, 28/02/2025
Dear Editor,
Thank you very much for the feedback provided and the opportunity to resubmit our manuscript (ijms-3474649) entitled “Role of soluble receptors cytokines in development and chemoresistance of gastric cancer”.
We would like to thank the reviewers for their careful reading of our manuscript. In order to keep the focus of our work, we revised the paper according to the reviewers.
We have addressed all the reviewers’ comments and the responses are appended below.
We hope that in the revised version our manuscript is suitable for publication on International Journal of Molecular Sciences.
Reviewer 3
Comments and Suggestions for Authors
- Should reword the following from the introduction as it is very confusing as is: “Incidence rates are highest in East Asia (Japan and Mongolia) and Eastern Europe and in general a direct correlation has been found between the incidence rate of gastric cancer and the country-level Human Development Index (HDI); index assessing four key dimensions of human development (Life expectancy at birth, mean years of schooling, expected years of schooling, GNI per capita)”
Thanks for your comments, we modified the text according to your suggestion.
- Bottom of section 2: “Cytokines may be a macro group of such biochemical markers that need more intensive research” comes out of left field and needs some introduction before making that claim. Or just remove it and talk about it later Done
- Page 4 paragraph 1: "Moreover, IL-10, IL-17 and TNFa represent markers of bad prognosis” does not have citations Done
- There is a large chunk of text in the first paragraph of 6.1 that does not have appropriate citations (sIL-6R intro information) Done
- Should figure 1 be showing an inhibitor? Inhibitors of soluble cytokines haven’t been discussed at this point. Text is “ leads us to hypothesize the role of these vesicles in the amplification of cytokine signaling through fusion with cells that do not normally express such receptors and through the same generation of their soluble receptors” but the figure doesn’t show that at all. Maybe this figure should go at the end? Thanks for your comment; we have modified the text, according to your suggestion.
5.5 - As I continued reading - it doesn't seem like inhibitors of soluble receptors are ever discussed in this paper, so I am unsure why the authors have included them in the figure. Done
- There is too much going on in the table - it feels like the text was just rewritten into the table. Should be condensed substantially to make more readable. Thanks for your comment; we have removed the table and simplified it in the text.
- The paper ends abruptly without any conclusions. I was expecting a section on inhibitors of soluble receptors or at least the impact of soluble receptors on chemoresistance and development of gastric cancer. The manuscript feels more like a review of the role of cytokines in these processes, not of the soluble receptors. An additional section or subsection directly addressing the role of the soluble receptors in gastric cancer is needed.
Thanks the review for your suggestion. We added the discussion and conclusion paragraphs in the text.
Comments on the Quality of English Language
There are substantial typos throughout as well as grammatical errors that impact the clarity of this paper. Thanks for your comments; we check the text with grammarly software
Response to the reviewer.
Thanks the reviewer for your suggestions. In order to satisfy your request we added the discussion and the text has been rewritten to improve the quality of the manuscript. Thanks the reviewer for comment. I hope the manuscript has improved enough to be considered publishable.
Kind regards,
Tiziana Notarangelo, Ph.D.,
Laboratory of Preclinical and Translational Research,
IRCCS-CROB, Referral Cancer Center of Basilicata,
85028, Rionero in Vulture, Italy
Tel. 0972 726239

Reviewer 4 Report
Comments and Suggestions for Authors
The authors touched on the important topic of gastric cancer. They focused on the role of soluble cytokine receptors in the development and drug resistance of gastric cancer.
The work is divided into sections, but some of them are too “heavy”. It would be useful to introduce more paragraphs, especially in the second section. Also, a figure summarizing this section would be useful.
The work is missing an ending. There is no summary or conclusion, the reader has the impression that there are missing pages in the paper.
Author Response
Rionero in Vulture, 28/02/2025
Dear Editor,
Thank you very much for the feedback provided and the opportunity to resubmit our manuscript (ijms-3474649) entitled “Role of soluble receptors cytokines in development and chemoresistance of gastric cancer”.
We would like to thank the reviewers for their careful reading of our manuscript. In order to keep the focus of our work, we revised the paper according to the reviewers.
We have addressed all the reviewers’ comments and the responses are appended below.
We hope that in the revised version our manuscript is suitable for publication on International Journal of Molecular Sciences.
Reviewer 4
Comments and Suggestions for Authors
The authors touched on the important topic of gastric cancer. They focused on the role of soluble cytokine receptors in the development and drug resistance of gastric cancer.
The work is divided into sections, but some of them are too “heavy”. It would be useful to introduce more paragraphs, especially in the second section. Also, a figure summarizing this section would be useful.
The work is missing an ending. There is no summary or conclusion, the reader has the impression that there are missing pages in the pap
Response to the reviewer.
Thanks the reviewer for your suggestions. In order to satisfy your request we added the discussion and the text has been rewritten to improve the quality of the manuscript. Thanks the reviewer for comment. I hope the manuscript has improved enough to be considered publishable.
Kind regards,
Tiziana Notarangelo, Ph.D.,
Laboratory of Preclinical and Translational Research,
IRCCS-CROB, Referral Cancer Center of Basilicata,
85028, Rionero in Vulture, Italy
Tel. 0972 726239
